# Corrosion Inhibition, Inhibitor Environments, and the Role of Machine Learning

**Anthony E. Hughes** [1,2,*], **David A. Winkler** [3,4,5], **James Carr** [6], **P. D. Lee** [7], **Y. S. Yang** [8], **Majid Laleh** [1,9] and **Mike Y. Tan** [9]

1 Institute for Frontier Materials, Deakin University, Waurn Ponds, Geelong 3216, Australia
2 CSIRO Minerals, Clayton, Melbourne 3168, Australia
3 Advanced Materials and Healthcare Technologies, School of Pharmacy, University of Nottingham, Nottingham NG7 2RD, UK
4 La Trobe Institute for Molecular Sciences, La Trobe University, Melbourne 3086, Australia
5 Monash Institute of Pharmaceutical Sciences, Monash University, Parkville 3052, Australia
6 School of Materials, University of Manchester, Manchester M13 9PL, UK
7 Department of Mechanical Engineering, University College London, London WC1E 7JE, UK
8 CSIRO Manufacturing, Clayton 3168, Australia
9 School of Engineering, Deakin University, Waurn Ponds, Geelong 3216, Australia
* Correspondence: tony.hughes@csiro.au

**Abstract:** Machine learning (ML) is providing a new design paradigm for many areas of technology, including corrosion inhibition. However, ML models require relatively large and diverse training sets to be most effective. This paper provides an overview of developments in corrosion inhibitor research, focussing on how corrosion performance data can be incorporated into machine learning and how large sets of inhibitor performance data that are suitable for training robust ML models can be developed through various corrosion inhibition testing approaches, especially high-throughput performance testing. It examines different types of environments where corrosion by-products and electrolytes operate, with a view to understanding how conventional inhibitor testing methods may be better designed, chosen, and applied to obtain the most useful performance data for inhibitors. The authors explore the role of modern characterisation techniques in defining corrosion chemistry in occluded structures (e.g., lap joints) and examine how corrosion inhibition databases generated by these techniques can be exemplified by recent developments. Finally, the authors briefly discuss how the effects of specific structures, alloy microstructures, leaching structures, and kinetics in paint films may be incorporated into machine learning strategies.

**Keywords:** machine learning; high-throughput testing; corrosion inhibition; localised corrosion; X-ray CT; data-constrained modelling

## 1. Introduction

The use of corrosion inhibitors in the modern context started in the 19th century and innovation in this field has tracked general advances in chemistry. Chromates, for example, were first used at the turn of the 19th century [1]. The development of new inhibitors was slow for most of the 20th century, due largely to the excellent performance of chromates and phosphates that saw them dominate the market [2]. However, the situation subsequently changed due to serious health and safety issues, especially for carcinogenic chromates [3]. Unfortunately, the very characteristics of chromate that make it a good inhibitor (strong oxidant and $Cr^{3+}/Cr^{6+}$ redox couple) also make it biologically active and toxic. Consequently, there has been a drive to find safer replacements. Traditionally, new inhibitors have been discovered using experimental measurements of corrosion inhibition by a particular chemical for a specific metal or alloy under specified environmental conditions. Thus, much of the data that has accumulated in the literature apply only to the performance

of inhibitors in relatively artificial environments, making it difficult to assess how they will perform in "real-world" situations. Moreover, modern manufacturing employs more multi-metal materials than in the past and the performance database does not address this issue.

Data analytics and machine learning (ML) are beginning to fill in this gap, albeit with a considerable lag compared with other related technologies. Figure 1 shows the approximate timelines for advances in high-throughput inhibitor assessment and computational inhibitor design over the past thirty years. In this period, there have been rapid advances in high-throughput (HTP) approaches for inhibitor discovery, new quantum mechanical methods, and ML. This paper reviews recent developments in these approaches. It explores the types of environments in which inhibitors must function, scoping out the types of electrolytes that inhibitors will encounter. The aim is to provide a rationale for the relevant collection of data for training ML models of small molecule corrosion inhibitors. It is intended to provide an overview of the challenges facing ML that arise from the complex environments in which inhibitors are required to operate. There is a focus on how 3D characterisation can inform this perspective.

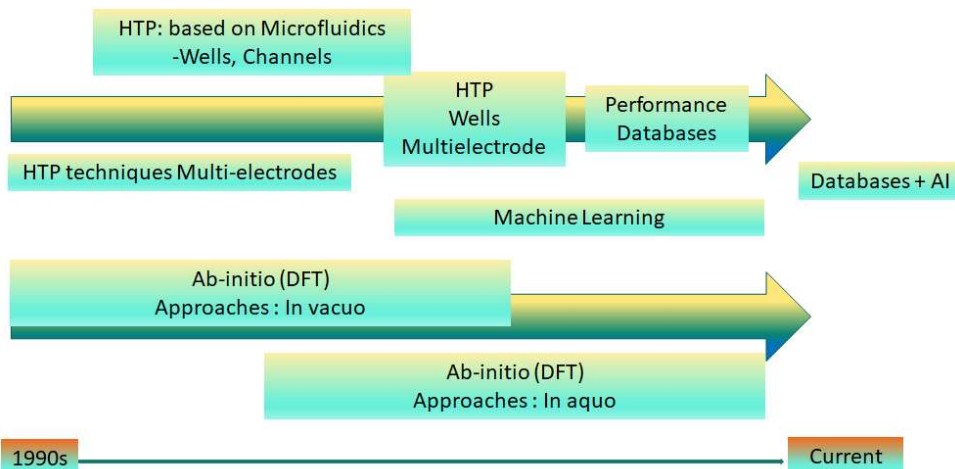

**Figure 1.** Timelines of progress in a range of approaches to the discovery of new inhibitors, including high-throughput (HTP) techniques; machine learning (ML) and artificial intelligence (AI); and ab initio approaches, such as density functional theory (DFT).

## 2. Recent Evolution of Inhibitor Assessment

Historically, the focus for inhibitor assessment was on mechanism- or performance-based studies of inhibitor performance [4–7]. Over the past two decades, there has been considerable focus on new techniques for measuring inhibitor performance rapidly (Figure 1, left). Traditional corrosion inhibition testing can take days to weeks to generate data, providing impetus to develop much faster high-throughput (HTP) tests. Several different HTP configurations were investigated in early work, such as different metal electrode configurations exposed to a corrosive solution and assessed via electrochemical measurements [8–13], microfluidic systems or microwells in which corrosion assessment is performed [14,15], and channels in which both chemical measurements and corrosion assessment were performed [16]. More recently, other HTP methods were developed such as using evolved hydrogen for screening corrosion inhibitors for Mg alloys [17]. Multipronged approaches that combine a range of different techniques were also proposed [18,19]. Table 1 provides a summary of these approaches. Initially, these HTP methods were developed to more rapidly identify inhibitors with promising performance via an experimental approach. However, over the last 10 years, it was recognised that they can generate large datasets for training ML models of inhibitor structure–property relationships that can be used to screen for, design, or optimise inhibitors [20].

In parallel with these high-throughput experimental developments, quantum mechanical (QM) modelling of inhibitor molecules and their interactions with surfaces has grown in capability and accuracy [21]. This has largely been driven by density functional theory (DFT) methods to model inhibitor molecular properties, interactions with surfaces, and performance. This approach is likely to be further enhanced by the outstanding developments in deep learning that allow for faster QM calculations by up to six orders of magnitude [22]. QM descriptors, such as electron affinity, ionisation potential, electronegativity, chemical potential, chemical hardness, and band gap, were used to characterise the inhibitors themselves [23–28]. Many initial studies were performed in vacuo and the descriptors derived from these calculations had little relevance to performance and very low predictive capabilities [25,27]. Even when water is considered, there are still questions about the relevance of the descriptors to the predictive power of these approaches [26,27]. However, subsequent studies of Mg corrosion suggest that the relevance of QM-derived descriptors may be metal-dependent [23]. Considerable improvement in DFT calculations occurred when aqueous environments were considered, and some excellent studies provided substantial mechanistic insight when the surface is also considered [28–31]. However, even these studies lacked sufficient predictive capability to screen large databases of candidate inhibitors and could not be deployed in other corrosion situations. Moreover, the resource requirements (largely computer time) of these QM methods are still a major limitation for modelling complex environments or large numbers of inhibitors. The reason for this is explored in more detail below.

**Table 1.** Types of high-throughput experiments and the method of assessment.

| Method | Method of Assessment | Application |
|---|---|---|
| Single-metal multi-electrodes | Corrosion current | Used in aqueous systems to assess a range of inhibitors |
| Wire beam electrode | Corrosion current | Used in electrolytes, but can also assess performances in structural configurations |
| Linear polarisation resistance | Resistance | Coatings |
| Multimetal electrodes | Corrosion current | Used for assessment as a function of pH and inhibitor concentration |
| Wells | Optical analysis of the level of corrosion | Multiple inhibitor assessment (individually or in combination) and pH on a single metal |
| Flow through channels | Optical analysis of the channel after testing and chemical analysis | Used to assess corrosion vs. time on single metal and multiple inhibitors limited by the number of channels |

In the last decade, ML approaches emerged as effective ways to model inhibitor performance in complex environments. It has major advantages over other approaches to inhibitor discovery [25,32]. First, it can, in principle, use almost all data in the literature, either conventional or HTP data. Second, it can employ a much wider range of descriptors (features) to represent the inhibitor molecules than can be extracted from QM calculations. Third, it has much better predictive capability than QM approaches, largely due to the use of more relevant descriptors and the ability to model complex phenomena without detailed knowledge of mechanisms. The number of inhibitor descriptors can be very large (>2000 in a recent study [25]), some of which are defined in Table 2. As an example, the topological distance between atoms X-Y and the frequency of occurrence of this separation means that the number of descriptors becomes rapidly larger as the number of atoms in an inhibitor increases. Indeed, the challenge in ML is to reduce the number of descriptors to a relevant number so that the learning process is not overfitted or compromised by the presence of descriptors with little or no relevance to the model (i.e., noise). In a recent study, some of the important descriptors identified from machine learning of 100 heterocyclic compounds were F03[N-S] and F04[N-S] topological descriptors for the frequency of nitrogen and

sulphur atoms that were three and four bonds apart and B02 [33] and F02(C-S) representing the presence/absence and frequency of occurrence of a carbon and sulphur two bonds apart. While these molecular descriptors are readily available from software packages such as Dragon, the use of arcane and relatively uninterpretable descriptors is falling out of favour. New descriptor families, such as molecular fingerprints, holograms, or signatures, are being more widely adopted because the relevant features from the model can be readily mapped back to exemplar molecules in the training set to provide guidance to organic chemists on how to improve inhibitors. Exciting developments on deep-learning-based generative methods (e.g., encoder–decoder networks, graph neural networks, and generative adversarial networks) also promise to suggest real, synthesisable molecules with desired inhibition properties [34,35].

**Table 2.** Typical molecular descriptors for quantum mechanical and machine learning approaches.

| Quantum Mechanical | Machine Learning |
| --- | --- |
| Highest occupied molecular orbital (HOMO) Lowest unoccupied molecular orbital (LUMO) Electron affinity Ionisation potential Electronegativity Chemical potential Chemical hardness Fundamental gap | A11 number of tertiary N atoms A31 number of secondary sulphur atoms Topological distances between atoms X and Y, e.g., B0n[X-Y], representing the presence/absence or frequency of occurrence of an X and Y n bonds apart Topological frequencies between atoms X and Y, i.e., F0n[X-Y] HOMT aromaticity index based on the length of the conjugated pathway C-XXX: numerous descriptors describing the number of different types of groups nCconj: number of non-aromatic conjugated C(sp2) groups nCp: terminal primary C(sp2) groups nTriazoles—number of triazoles nBenzene—number of benzene rings |

As ML methods are data-driven, the main challenge is the generation of large databases of inhibitor performance, both as repositories of knowledge and for training ML algorithms. The larger and more chemically diverse the training data, the more generally applicable the ML model will be in predicting properties of new molecules in, for example, a virtual database. Models work best when they predict the properties of molecules in or near their domains of applicability (the region of chemical space in which they are trained). The predictive power of an ML model is broadened by a larger and more chemically diverse training dataset. One issue with the general literature is that, for the most part, the performance of good inhibitors is reported and there is much less information about poorer inhibitors. ML models are most robust when trained on a wide range of inhibitor performance values; therefore, the properties of poor inhibitors are also important for training ML models. Here, HTP experiments can fill this gap by rapidly generating large data sets that contain a wide range of inhibitor performances.

## 3. Environments in Which Inhibitors Typically Perform

Before discussing the requirements of large databases for inhibitor performance, it is important to examine the types of environments in which inhibitors are required to perform, as this may inform the types of HT experiments required for populating databases.

One drawback of conventional inhibitor assessments (ranging from industry tests to laboratory tests) and even the newer HTP methods is that they are largely performed under conditions or in electrolytes (typically neutral NaCl) that are not relevant to the operating environment of the inhibitor in "real-world" applications. Another disadvantage is that the conditions of exposure remain the same throughout the test, i.e., "constant stress" testing is used. Constant stress is rarely experienced in real-world applications, where structures to be protected are often exposed to diurnal cycles, temperature cycles, wet/dry cycles, tidal cycles, mechanical cycles, and chemical changes in the environment. These conditions are often referred to as "cyclic stress". In industry, a valid, accelerated test design that

reflects these cyclic stresses has been the source of debate for many years [36–39]. Even in laboratory tests, time variation is important [40]. However, it is not surprising that most corrosion inhibition literature uses a limited number of electrolytes since this makes it easier to compare studies. Almost invariably, little attention is paid to the underlying structure to be protected or the delivery system for the inhibitor, i.e., coating systems. In this section, the focus is on providing examples of common structures (e.g., riveted joints) and common environments in which inhibitors are expected to perform. The intention is not to provide an exhaustive review of these topics, but to outline what types of testing might provide useful metadata for training improved ML models.

Dividing this topic into categories is not straightforward since it could be organised in a range of ways, such as structure type, corrosion type, or environment (noting that there are international standards that classify environments, such as ISO 12944). There are also many examples that deal with the corrosion of structures in particular environments [41–47]. In addition to these typical industrial environments, there are also areas such as under biofilms or in vitro applications for implants [48], which, because of their broad scope, are not addressed here. Regarding corrosion types include pitting corrosion; crevice corrosion; exfoliation corrosion; and undercoating types of corrosion, such as filiform corrosion and under-paint delamination. The main question is whether these classifications provide appropriate frameworks for designing tests that provide relevant data for training robust and predictive ML models. For example, a classification system could be based on combinations of the above categories, e.g., lap-joint/marine environment/inhibited epoxy for bridges near the sea or lap-joint/severe marine environment/inhibited epoxy for offshore wind turbines or oil wells. When corrosion occurs in any of these structures, electrolytes will develop that will be specific to the environment and the structure, and localised corrosion will develop where differential oxygen supply leads to the development of anodes and cathodes. The effectiveness of any corrosion inhibitor in these electrolytes will depend on the composition of the electrolyte. For example, Figure 2 shows a riveted joint from an aircraft. Clearly, there is an occluded volume with an opening to the external environment that begins with a fracture through the paint system, then penetrates down through the joint between the fastener and the fastened metal. The electrolyte in this region will have entirely different characteristics from those in standard tests. It will contain metal ions from the corrosion of the fastened metal, ions from dissolved deposited salts, and inhibitor ions from the primer. This complex electrolyte may accelerate or slow the progress of corrosion. The corrosion may progress to underfilm corrosion, then to filiforms or delamination. Alternatively, it may result in an attack on the metal microstructure, leading to exfoliation corrosion. The electrolytes that develop in all these cases will be different. Systematic studies of inhibitor performance under real-world conditions are lacking but would provide valuable additional performance data for ML.

In coating systems, performance can be categorised by its barrier or inhibitor delivery properties. It is worth noting that coatings are not often applied as a standalone product. Usually, there is some sort of metal preparation, primer application (which may be used to promote the adhesion of a topcoat or deliver an inhibitor), and one or more overcoats to provide a barrier and/or cosmetic properties [49–53]. Cooling water systems are another area where corrosion occurs in an occluded space, in this case, underscale corrosion [54].

In all cases, the loss of performance is related to the transport of water and electrolytes into and out of the coating (unless there is mechanical damage from the outset). Thus, techniques that can explore the 3D structure directly are emerging as important. One such approach is X-ray computed tomography (CT), which is being used to study inhibitor distributions (and other organic phases) in coating systems to elucidate internal inhibitor particle distributions. For example, Figure 3 summarises the spatial relationship between inhibitor particles, where clusters are formed in a model system containing only strontium chromate ($SrCrO_4$) particles. Figure 3a shows the two largest clusters (orange and yellow) intermingled within the distribution of other inhibitor particles, all of which are embedded within an epoxy binder. Evidence for these extended clusters of inhibitor particles

is reported in several studies [55–58]. In this example, the clusters have a range of sizes (Figure 3b). The largest cluster contains 3.4 μM of SrCrO₄, which, at a critical concentration of 10 μmol/cc [59], which is enough to reach the critical inhibitor concentration to dose one-third of a cubic centimetre. Conspicuously, X-ray CT not only provides information about the amount of chromate (or any other inhibitor) but also its distribution and potential transport pathways since it was shown that chromate can only escape to the external electrolyte through channels created by its dissolution. For ML models, the kinetics of release for inhibitors from this channel network is as important as the intrinsic inhibitor efficiency since it determines the dose of the inhibitor that can be delivered to the metal surface. The dose needs to be above the critical inhibitor concentration for effective inhibition.

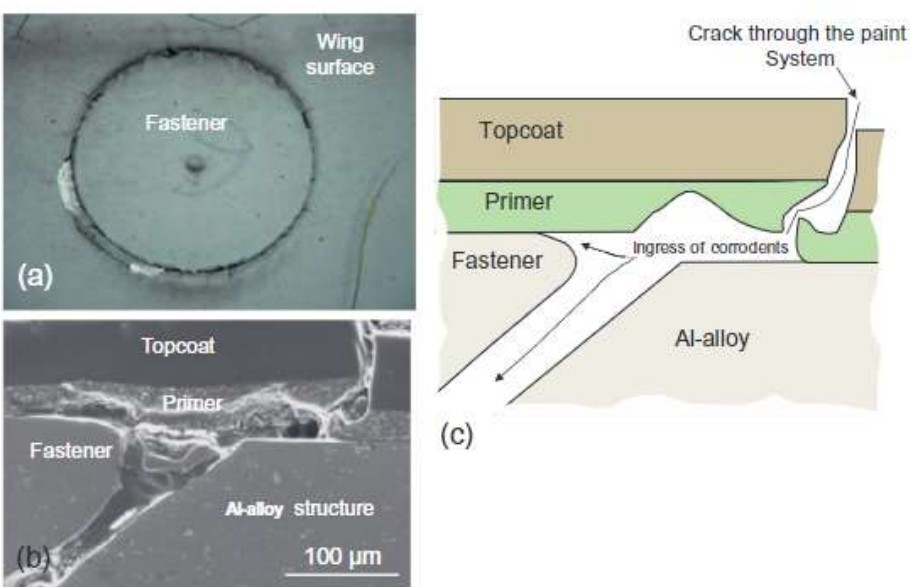

**Figure 2.** (**a**) Crack in the protective coating around a fastener from an aircraft wing; (**b**) cross-section of the paint coating system, fastener, and metal fixed by the fastener; and (**c**) schematic of (**b**) delineating the path of ingress of corrodents into the fastener cavity. Reprinted with permission from [49].

A separate study examined the role of second-phase particles in the creation of a channel network. Specifically, the development of transport pathways through the inclusion of calcium sulphate (CaSO₄) was examined, along with the SrCrO₄ particles in the coating formulation. Figure 3d shows the distribution of CaSO₄ and SrCrO₄ particles embedded in epoxy in an X-ray CT slice. A defect was scribed into the edge of the material and the sample was then exposed to an electrolyte. It can be seen (Figure 3) that the relatively soluble CaSO₄ particles dissolved at the defect site creating voids, whereas the sparingly soluble SrCrO₄ showed evidence of a much lower level of dissolution, which was evident for the larger particles where there was only a minor decrease in particle size around their perimeter. It was shown, in some instances, that the voids created by the dissolution of CaSO₄ were connected to voids around the SrCrO₄.

While these studies highlight channel network formation, the key point from the perspective of this paper is that the electrolyte in these channels (dissolved inhibitor and other ions) is complex, particularly since it must eventually mix with the external electrolyte or even react with other particles in the coating. For example, Kopeć et al. [57] observed the reaction of chromate with the surface of barium sulfate (BaSO₄) particles, indicating that reactions within channels at particle surfaces are important for determining the amount of inhibitor that can be delivered to a defect. The inhibitor remaining in the electrolyte that develops around dissolving particles *within* the epoxy eventually becomes mixed with the external electrolyte, creating some extremely complex and concentrated electrolytes with a major influence on inhibitor performance. These electrolytes need to be

well characterised and inhibitor performance databases need to be developed to reflect the inhibitor efficiencies.

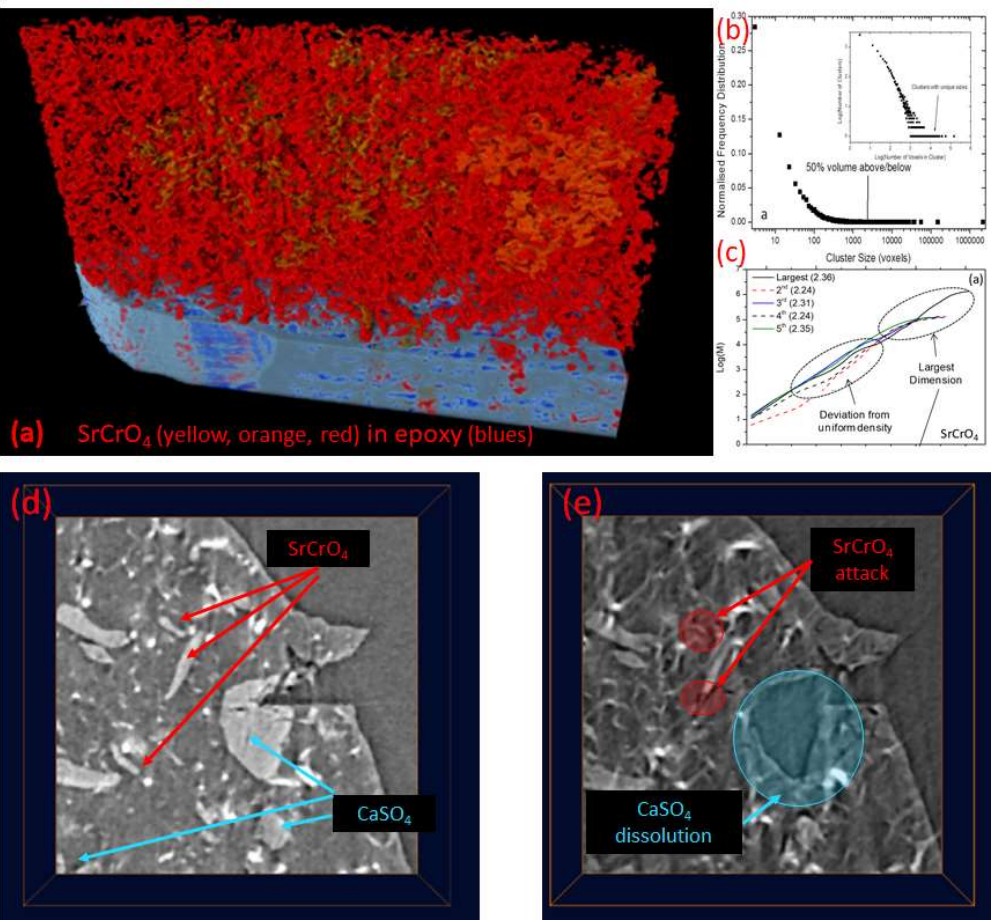

**Figure 3.** (**a**) Example of chromate clusters in yellow and orange among the rest of the SrCrO$_4$ particles in a piece with model coatings containing only SrCrO$_4$ particles and epoxy. (**b**) Cluster distribution size distribution and (**c**) cluster volume versus characteristic length, which yields the cluster fractal dimension. (**d**) Before and (**e**) after leaching CT slices of a defect in a model coating containing CaSO$_4$ and SrCrO$_4$ particles in an epoxy. Parts (**b**) and (**c**) were reproduced with permission from [60].

With respect to the barrier properties of coatings, mechanical damage is the most important pathway for the ingress of corrodents. However, mechanical damage also provides insight into how inhibitor-containing electrolytes may be transported through the material. For example, Ranade et al. [61,62] used finite element analysis to investigate strain around particles. They identified elevated strain regions forming around the inorganic particles that can merge to form connected structures. Figure 4A shows a cutaway of the strain distribution around several particles (blue and green) embedded in a polymer. Figure 4B shows a 3D reconstruction of a strain iso-surface around the particles, highlighting the interconnection of the strain iso-surface between particles and the potential for void formation. X-ray CT studies of the same system in Figure 4C show that the polymer around particles may have a different density to polymer elsewhere in the sample (N.B.: it is a difference in the linear absorption coefficient, which is interpreted as a difference in polymer density). Apart from the red, which represents the particles, all other colours represent differences in polymer density, with blue representing the lowest density. The relationship is highlighted in Figure 4D where only the low density (blue) and particles (red) are shown. While cause and effect are not suggested here, the co-incidence of the von Mises strain distribution with a lower-density polymer component highlights how strain could interact with the coating microstructure to enhance the development of transport

networks within a coating. This phenomenon was further investigated by Laird et al. [58], who showed that this type of interaction may be responsible for microcrack formation in coatings.

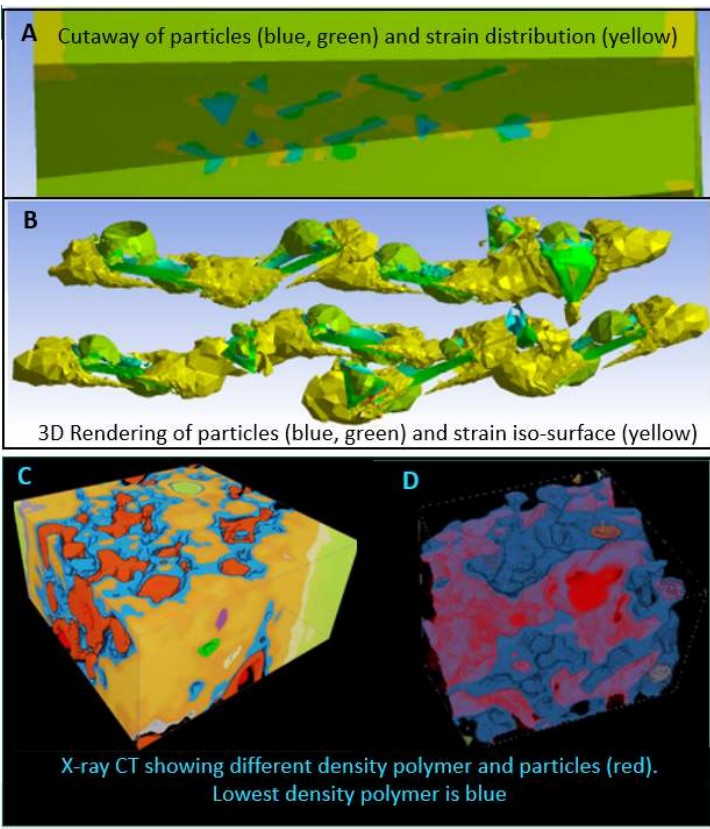

**Figure 4.** (**A**) Cutaway of an FEA of a particular arrangement of particles in a polymer matrix, (**B**) 3D rendering of a strain iso-surface (yellow) between particles (green and blue), (**C**) X-ray CT showing the density variation of polymer around particles (red), and (**D**) low-density polymer distribution around particles. Apated from Ranade et al. [58] and reproduced with permission.

Employing a standard laboratory-based method to evaluate the efficiency of an inhibitor for oilfield corrosion has always been a challenge mainly due to the difficulties in simulating the significant variables in the real application conditions, including temperature, pressure, flow, and compositions (solids, liquids, and gases). The most generally accepted laboratory methodologies for evaluating corrosion inhibitors in oil and gas industries were described in some ASTM documents [63–68], which include flow loop, rotating cylinder electrode, rotating cage, wheel test, and jet impingement. Apart from selecting the most efficient inhibitor for a particular application using any of the methods mentioned earlier, the users also need to consider the safety and environmental requirements, such as compatibility with other chemicals, which adds to the complexity of developing new corrosion inhibitors. ML can utilise data in the literature to speed up the discovery of corrosion inhibitors with acceptable accuracy. The greatest challenges faced when using this approach are perhaps finding relevant, consistent, and reliable data.

## 4. Localised Corrosion Environments and Attack

Thus far, this paper has focused on structures within coatings through which inhibitor ions travel and potentially mix with other ions in the electrolyte transport network to react with surfaces exposed to the transport network. They eventually mix with the external electrolytes that are dominated by the local environment, particularly the electrochemical environment, when they emerge from the primer. This environment is influenced

by local atmospheric, tidal (for structures at the waterline), and operating conditions (e.g., in machinery); occluded spaces (e.g., under insulation); fastened joints; etc. Within these environments, the electrochemistry will dictate the electrolyte compositions that will range from acidic (where anodic reactions occur, and metal ions accumulate) to alkaline (where cathodic reactions occur and precipitates may develop, which also change the electrolyte composition).

At the microscopic level, corrosion is commonly classified as pitting corrosion, exfoliation corrosion, filiform corrosion, underfilm corrosion, and an intergranular attack. The electrolyte environment in most of these cases depends on the local electrochemistry, the inhibitor type and concentration, and the metal under attack. Electrochemistry divides these regions into alkaline and acidic electrolytes, which have different chemistries. The inhibitor will change the electrolyte because of its presence, i.e., alter the ion concentration or speciation, but also because it may precipitate metal ions in the electrolyte, and thus, also change its composition or even create porous overlayers where corrosion can occur as described below. With respect to the underlying metal, complex effects that occur during metallic dissolution at anodic sites may see preferential dissolution into the anolyte solution (electrolyte at the anode) and accumulation of other elements on the surface of the metal. Cu accumulation in Al alloys [69–73] and Ni accumulation under the passive layer in austenitic stainless steels [74,75] are examples.

Probing the chemistry and pH at these sites is not straightforward and depends on the size of the structure to be probed. Crevices can be microscopic, but many macroscopic structural components that contain occluded volumes (e.g., lap joints and riveted structures) are also ideal sites for crevice corrosion. In these types of structures, and for localised corrosion more generally, there is a serious lack of corrosion inhibition data. Localised corrosion inhibition is a major and difficult issue in corrosion prevention [76,77].

There are numerous methods for observing and characterising localised corrosion, as summarised below. Many of these can provide inhibitor performance data and it should be remembered that inhibition efficiency is often derived from localised corrosion currents and there is the opportunity to use these directly, along with some other metrics described below. The cyclic polarisation method, such as the standard ASTM G61 [78], is probably the only standardised, traditional electrochemical method that can be used to determine the relative localised corrosion susceptibility. It involves the anodic polarisation of a specimen until localised corrosion initiates, as indicated by a large increase in the applied current. An indication of the susceptibility to initiation of localised corrosion in this test method is given by the potential at which the anodic current increases rapidly, i.e., the breakdown potential. The nobler (more positive) this potential, the less susceptible the alloy is to initiating localised corrosion. The conventional understanding is that the breakdown potential is the potential above which pits are initiated, while the repassivation potential obtained using a reverse scan is the potential below which pits repassivate. There were also numerous attempts to use the amount of hysteresis in the cyclic scan as a measure of localised corrosion susceptibility, with varying degrees of success. It should be noted that the results of the cyclic polarisation test are not intended to correlate in a quantitative manner with the rate of localised corrosion under natural open circuit corrosion conditions. Another issue is that these test methods can only provide one set of test results (e.g., a polarisation curve) because the test specimen would be destroyed by such a passivation measurement.

During the past two decades, the development of advanced physical and scanning electrochemical techniques, such as atomic force microscopy (AFM), scanning Kelvin probe (SKP), scanning Kelvin probe force microscopy (SKPFM), scanning reference electrode technique (SRET), scanning vibrating electrode technique (SVET), local electrochemical impedance spectroscopy, and scanning electrochemical microscopy, has facilitated substantial progress in research on localised corrosion and its inhibition. These test methods can provide substantially more corrosion data by repeatedly scanning corroding metal surfaces for ML and the modelling of localised corrosion systems. SKP and SKPFM are scanning probe techniques (SPT) that permit mapping of the topography and Volta potential distri-

bution on electrode surfaces. They scan the electric potential just above the electrolyte over an electrode surface in order to detect Volta potential differences over different parts of the electrode. SKPFM combines an SKP with AFM and uses smaller probes that operate at much smaller distances from the surface. Thus, SKPFM has an improved lateral resolution (<0.1 μm) compared with a classical SKP of 100 μm. The SKPFM technique provides both Volta potential and topographical data with submicrometre resolution. Williams et al. [79] used SKP to study the influence of trivalent cerium cations on the kinetics and mechanism of corrosion-driven delamination processes affecting polyvinyl butyral coatings adherent to the intact zinc surface of hot dip galvanised steel. McMurray et al. [80] employed an SKP probe to study the effects of incorporating dispersions of $SrCrO_4$, silica, and bentonite-based cerium (III) cation exchange pigments within the polyvinyl butyral coating in the absence and presence of a mixed oxide/chromate rinse surface pretreatment. There appeared to be a synergy between the inhibiting pigments and the chromate rinse pretreatment. This synergy is believed to be related either to the ease with which a cerium hydroxide/oxide inhibition film can form on the oxide-covered surface of the pretreated material or to the underfilm interaction of pigment-derived cathodic ($Ce^{3+}$) and pretreatment-derived anodic ($CrO_4^{2-}$) inhibitors. Yasakau et al. [81] studied the corrosion protection mechanism of aluminium alloy (AA) 2024-T3 via Ce and La inhibitors in chloride media using SKPFM, AFM and SEM coupled with energy dispersive spectroscopy. Intermetallics (IMs) and their compositions in AAs were identified using conventional SEM and EDS. SKPFM was used to measure IM-related Volta potentials. Employing these high-resolution and in situ techniques provided a deep understanding of the details of the physical chemistry and mechanisms of the corrosion processes. REM-based inhibitors showed sufficient suppression of the localised corrosion processes, especially in the case of pitting formation located around the S-phase IM particles. SPTs that scan and detect local electrode potentials, galvanic currents, and local electrochemical impedances at the metal surface or metal-electrolyte interface were used to investigate localised corrosion. SRET and SVET [82,83] are designed to probe local ionic currents flowing in the electrolyte by detecting small potential variations over electrode surfaces where local electrode processes occur. In SRET, this is usually achieved by scanning a passive reference probe parallel and near the metal surface and keeping the distance constant by monitoring the *IR* drop between the two. Thus, potential variations caused by ionic current flows within the electrolyte can be measured if the probe is near corrosion sites and if the electrolyte conductivity is not too high. Yasakau et al. [84] used SVET to investigate the potential interaction of corrosion inhibitors, including cerium nitrate ($Ce(NO_3)_3$), with components of the sol–gel films and their effects on the corrosion protection of AA2024. The results demonstrate that $Ce(NO_3)_3$ did not affect the stability of sol–gel films and conferred additional active corrosion protection. Montemor et al. [85] investigated the effects of $Ce(NO_3)_3$ on the pre-treatment of an AZ31 Mg alloy using SVET in conjunction with conventional potentiodynamic polarisation and open-circuit potential measurements. They found that the pre-treatment reduced the corrosion activity of the AZ31 Mg alloy in solutions containing chloride ions. They also investigated the surface composition using X-ray photoelectron spectroscopy and Auger electron spectroscopy. They confirmed the presence of a surface film containing the rare-earth cation and that the composition dependent on the immersion time in the case of the cerium pretreatment. Each SPT has its advantages and limitations, thus different techniques are often combined and applied in a synergistic manner. For instance, traditional optical microscopy, SEM, and EDS techniques are often combined with scanning probe techniques to provide topographical and chemical information that is often critical for corrosion inhibitor research.

Unfortunately, the SPTs described above could not be used to acquire corrosion data from some corrosion systems, for instance, SPTs cannot scan corrosion under a coating/inhibitor film, under solid deposits, or in crevices. It is therefore necessary to design tests to effectively measure localised corrosion inhibition in order to generate data for ML. Underfilm corrosion is one type of corrosion that occurs in many environments [86]. This is exemplified by a practical example of developing a test for evaluating the performance

of under-deposit carbon dioxide corrosion inhibitors under simulated oil flowline conditions [87]. Although there are several ASTM standard testing methods for evaluating oilfield corrosion inhibitors in the laboratory (e.g., [63–66]) and many NACE papers that discuss results from this type of testing, there are few methods that can be used to evaluate inhibitor performance for a more complex form of corrosion, such as corrosion under deposits. Corrosion inhibitors are used to prevent oil pipeline failure due to pitting and mesa corrosion under solid deposits, such as sand and biofilms. A problem is that the efficiency of corrosion inhibitors is often unknown because their assessment using normal corrosion testing techniques [88] is considered intractable. Under-deposit carbon dioxide corrosion is believed to be controlled by factors including galvanic effects between a large cathode (pipeline surface) and a small anode (surface under deposits), failure of inhibitors to penetrate the deposits, and the retention of aggressive species within the deposits. A corrosion inhibitor test should effectively simulate these controlling factors and measure their effects on corrosion rates and patterns. Several different test methods were designed in an attempt to simulate these complex environmental factors. For instance, de Reus et al. [89] used two sets of three electrode arrays, with one set covered with sand and another directly exposed to a brine solution. Such a setup was designed to allow for simultaneous electrochemical measurements at both uncovered and covered areas for direct electrochemical comparison. This method should be able to detect the effects of localised differential concentration cells, the possible failure of inhibitors to penetrate the deposits, and the possible retention of aggressive species in the deposits on corrosion rates. However, it is unable to simulate galvanic corrosion activities associated with under-deposit corrosion mechanisms. Clearly, failure to measure galvanic currents flowing between covered and uncovered areas would lead to an underestimation of under-deposit corrosion. Pedersen et al. [90] designed a test device consisting of three specimens, with two specimens covered with sand and one directly exposed to a brine solution. One of the covered specimens was coupled to the uncovered specimen. Corrosion inhibitors were assessed by detecting galvanic currents flowing between the sand-covered and uncovered specimens. This test allowed for the simulation of galvanic activities between a large cathode and a small anode and the measurement of galvanic currents. This device showed that the sand-covered specimen was anodically polarised and was under localised corrosion attack. However, this test may not be suitable for highly resistive media, such as multi-phase fluids, where a high electrolyte resistance prevents galvanic current flow between the sand-covered and non-covered specimens. Another issue is that the device would not effectively simulate the localised chemical changes over a partially covered metal surface due to local corrosion reactions and the retention of aggressive species in the deposits. These test methods are able to generate useful data for ML and the modelling of under-deposit corrosion and inhibitors; however, these data usually only contain information about corrosion rates and do not contain information about the corrosion distribution and pattern.

Tan et al. [87] designed a test method based on a wire beam electrode (WBE) that not only avoids difficulties in the testing and evaluation of under-deposit inhibitors but also has the possibility of generating site-specific data, thereby defining the corrosion distribution pattern, which would be useful for ML. The WBE method is able to repeatedly measure and map the corrosion and inhibition processes under a solid deposit or under a coating film, providing a means of high-throughput performance testing and generating lots of localised corrosion and inhibition data for ML. Figure 5 shows photos and a schematic diagram of the apparatus in which the WBE working surface was partially covered with a rubber "O" shaped ring filled with sand to simulate a localised under-deposit corrosion environment. In a typical localised corrosion inhibitor test [87], the WBE was made from 100 mild steel wires embedded in epoxy resin that were insulated from each other by a very thin epoxy layer. Each wire had a diameter of 0.19 cm and acted both as a mini-electrode (sensor) and as a corrosion substrate. During $CO_2$ corrosion testing, $CO_2$ sparging was continued to maintain a virtually oxygen free environment. During corrosion exposure periods, all wire terminals of the WBE were connected and electrons could move freely

between wires, akin to the case of a larger one-piece electrode. Under-deposit corrosion processes were monitored by mapping galvanic currents across the multi-electrode array to understand how localised corrosion initiated and propagated under sand, and how it changed with the introduction of the inhibitor [87].

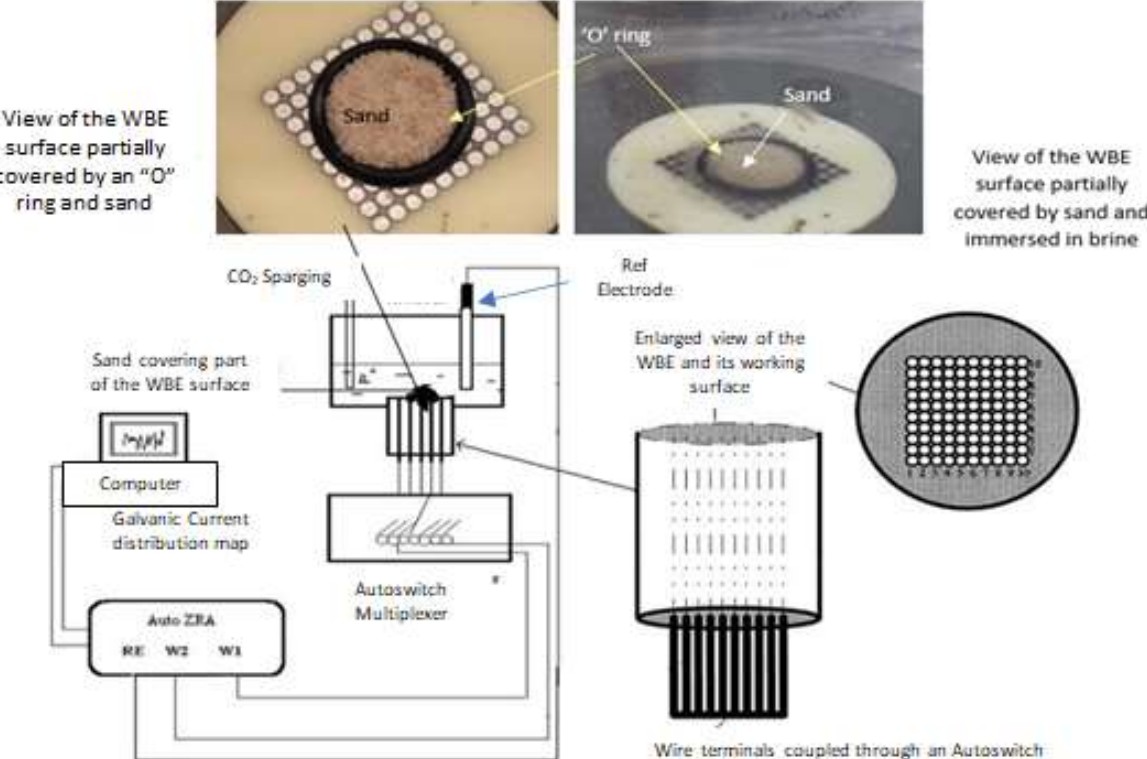

**Figure 5.** Schematic diagrams showing a WBE test setup for under-deposit corrosion and its inhibition [87]. Reproduced with permission.

Very different corrosion behaviour was observed for a partially covered WBE surface exposed to a $CO_2$-saturated brine environment with and without the presence of the corrosion inhibitor imidazoline. Without the inhibitor (Figure 6a), positive galvanic currents concentrated mainly on areas uncovered by the "O" ring and sand, while cathodic currents were distributed mainly over areas closer to the $CO_2$ sparging tube. This result suggests that under-deposit corrosion was not occurring in a $CO_2$-saturated pure brine solution under ambient temperature because the area under the sand behaved as a cathode. When imidazoline was added to the brine solution, the corrosion anodes and cathodes quickly changed locations (Figure 6b). Corrosion anodes shifted to areas covered by the "O" ring and sand, while cathodes were located mainly over the four corners where no sand was present. This is surprising and interesting since this result suggests that the addition of an inhibitor resulted in the formation of a corrosion anode under the sand deposit. This experiment illustrates the complexity of under-deposit corrosion and the effects of inhibitors, as well as some of the complications of generating data for ML. Corrosion potential distribution maps confirmed that the addition of the inhibitor imidazoline significantly changed the corrosion potential and its distribution over the WBE surface. The potential difference between the cathodic and anodic areas was >250 mV for sites where the inhibitor could easily penetrate while acting as the cathode and areas buried under the deposit were acting as the anode, initiating under-deposit corrosion. An increase in imidazoline concentration was found to reduce overall corrosion rates but enhance localised corrosion. This result was confirmed by weight-loss coupon tests [87].

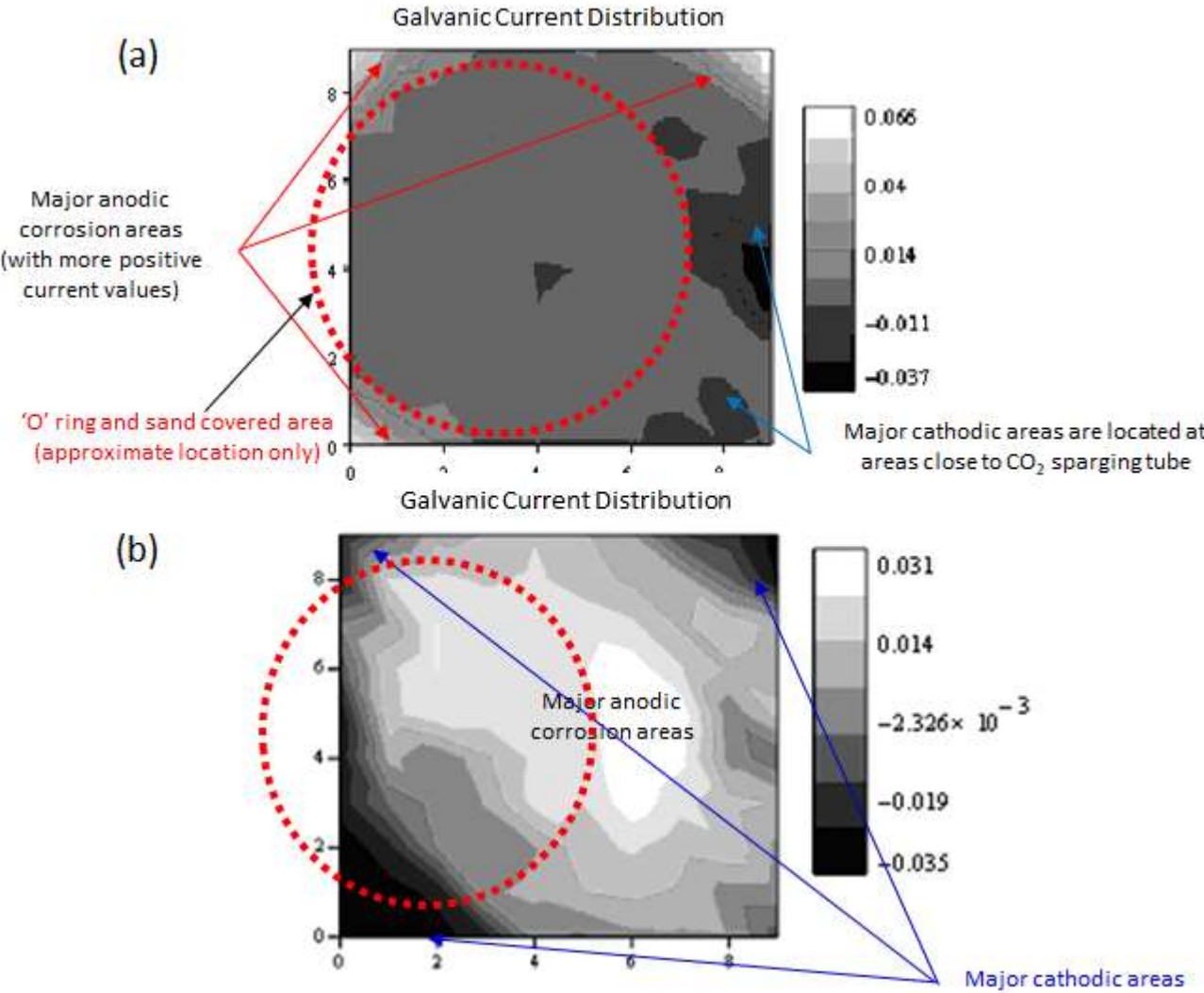

**Figure 6.** (**a**) Galvanic current distribution (in mA/cm$^2$) maps measured over a WBE exposed to a brine–CO$_2$ corrosion environment for 18 h without an inhibitor present, and (**b**) 97 min after 30 ppm of the inhibitor imidazoline was added into the system [87]. Reproduced with permission.

The WBE method was also used to search for new types of localised corrosion inhibitors. In a typical study [87], the WBE was used to investigate the behaviour of imidazoline and acid-functionalised resorcinarene as steel corrosion inhibitors in CO$_2$-saturated brine solutions. Both imidazoline and resorcinarene acid provided excellent inhibition to general CO$_2$ corrosion; however, imidazoline was found to aggravate localised corrosion by creating a small number of major anodes that were concentrated on a small area of the WBE surface, leading to highly concentrated anodic dissolution. The resorcinarene acid showed distinctively different behaviour by generating many minor anodes that were randomly distributed over the WBE surface, leading to negligible general anodic dissolution. These results indicate that resorcinarene acid provides effective localised corrosion inhibition by promoting a random distribution of insignificant anodic currents [87]. Muster et al. [11] described a rapid screening approach for the evaluation of candidate corrosion inhibitors, including REM compounds. García et al. [91] used this multi-electrode system to study the influence of pH on corrosion inhibitor selection for AA2024-T3. They performed tests in 0.1 M NaCl solution containing potassium dichromate, cerium dibutylphosphate, and cerium chloride. These results demonstrate the applicability of using the WBE method to acquire corrosion data from more complex and localised corrosion systems, for instance, corrosion under a coating/inhibitor film, under solid deposits, or in crevices.

In terms of localised corrosion, research into the morphology of attack was aided by the development and refinement of several approaches that give a better 3D understanding. These studies help to refine an overview of the physical confinement that aggressive electrolytes experience as they develop. For example, CT and X-ray synchrotron techniques emerged as useful methods for probing the internal structure of pits [92–99]. They were used to identify pit growth and salt films in steels [93].

Figure 7 shows a corrosion front moving through an additively manufactured SS316L microstructure. Figure 7a shows the corrosion front in buff colour and lack-of-fusion (LOF) pore structures ahead of the front. The metal was coloured a transparent blue so that the LOF structures can be seen. Typical LOF structures are indicated by the black arrows in Figure 7a. Of particular interest is the material immediately behind the pit front that is probably a concentrated iron-containing chloride electrolyte (in red and indicated by red arrows in Figure 7a).

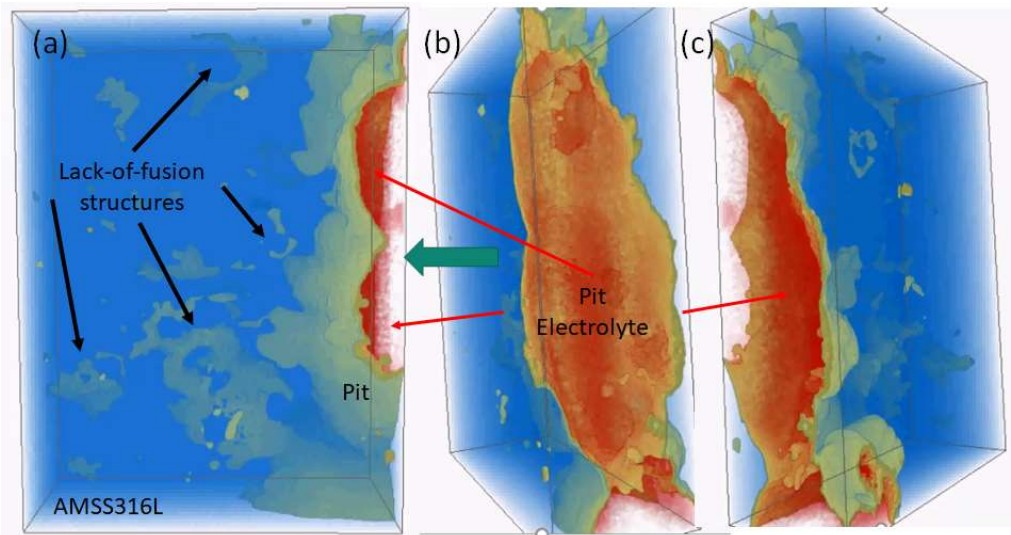

**Figure 7.** X-ray CT reconstruction of pitting produced in FeCl₃·6H₂O moving through an additively manufactured SS316L alloy. The steel is coloured in blue but has been made transparent so that the internal structures can be observed. The buff features represent both the corrosion front (on the left in (**a**)) and lack-of-fusion pore structures (several indicated using black arrows). (**b**) Rotated view relative to (**a**), showing the interior of the pit. The green arrow indicates the direction in which the viewer is looking into the pit. The red qualitatively reflects the distribution of concentrated corrosion product, which is mainly but not purely, ferric chloride (probably gel-like). (**c**) Further rotation that provides another aspect of the pit and internal corrosion product.

This can be seen more clearly in Figure 7b,c when looking into the pit from the direction in which the steel was already corroded (the green arrow indicates this direction). The intensity of the red (also transparent) varies within the pit, reflecting both the depth through which it is viewed, as well as the concentration. The concentration cannot be obtained from this image but can be obtained from further analysis of the X-ray CT data and by sectioning and direct analysis. In stainless steels, the progression of corrosion in these types of pits is determined via salt film formation, dissipation, and passivation [100,101]. A combination of X-ray CT and DCM can provide qualitative information on the composition and distribution of corrosion products and protective layers within the pit.

The anolyte (electrolyte at the anode) plays a key role in pitting kinetics in localised corrosion. Generally, in chloride solutions, the anolyte is a concentrated, acidic metal chloride solution. Salt film formation was studied extensively in ferrous metals (Table 3), showing that it generally dissipates when it is below 30% of its saturation level, causing repassivation [102]. The addition of other anions influences the formation and dissipation of this solution. For example, studies showed that sulphate ions influence the pitting

processes through modification of the anolyte solution [103–105]. Higher valency anions preferentially accumulate in the anolyte solution compared with the monovalent chloride ion [104], and if these anions have a significant effect on the stability of salt films or the kinetics of dissolution, then they can impart a degree of inhibition [105]. While some electrochemical and wet chemical studies performed on chloride and sulphate mixtures were reported, particularly for ferrous metals, most studies generally focus on a simple NaCl electrolyte in the neutral pH range. Clearly, the high pH of the catholyte (electrolyte at the cathode) and low pH of the anolyte solutions differs greatly from the near-neutral conditions, as does the speciation where the majority of the standard testing is performed for inhibitor efficiency. Moreover, electrolytes generated under real-world conditions are much more complex, with many more cations and anions compared with simple salt solutions, such as NaCl [106–109].

**Table 3.** Some electrolyte, corrosion product, and salt film compositions.

| Metal | Pit Electrolyte | Reference |
|---|---|---|
| Ni | $NiCl_2 \cdot 6H_2O$ salt film/sat electrolyte | [92,110] |
| Fe | $FeCl_2 \cdot 6H_2O$ salt film/sat electrolyte | [92,111] |
| SS316L | $FeCl_2 \cdot 6H_2O$ salt film/sat electrolyte | [92] |
| SS304 | $FeCl_2$, $FeSO_4$ (probably with waters of hydration) | [105] |
| Steel bar (reo) | Anolyte $FeCl_2$ (pH 2.7), catholyte NaOH (pH 13) | [112] |
| AM30 (Mg alloy) | $MgCl_2 \cdot MgOHCl$ [1] | [113] |
| Fe | $4.25 \pm 0.05$ M for $FeCl_2$ | [111] |
| Austenitic SS | 3.5 M $Fe^{2+}$, 1.1 M $Cr^{3+}$, and 0.5 M $Ni^{2+}$ | [114] |
| AA2024 | >20 wt% $Cl^-$ and $Al(OH)_3$ in a filiform head | [115] |
| AA2024 | Strong chloride environment with O and Al in pits for filiform | [94,116] |
| AlCuMg$_2$ | Al-hydroxide gel with Mg, chloride, and Cu ions under seawater droplet (S-phase embedded in AA2024-T3) | [117,118] |

[1] The composition for the hydroxychloride was modified to MgOHCl compared with that posted by the authors, which was $Mg(OH)_2Cl_2$.

Stable pitting in metals is not only dictated by the composition of the electrolyte but also depends on the concentration difference between the anolyte and the external electrolyte and the ability to establish a long enough diffusion path to maintain a concentration gradient [104]. This is commonly summarised as the pit stability product (i.r$_{pit}$) being greater than $10^{-2}$ Acm for several metals [103,104,119–122]. For example, according to Cheng et al. [120] in SS304 the concentration difference between the anolyte and bulk solution (with no metal ions) must be maintained above 3 mol/L and the upper limit required the concentration to be <4 mol/L on the basis of salt solubility [111]. These values reflect the necessity to maintain a sufficiently large concentration difference, in combination with a long enough diffusion path, for the anolyte solution chemistry to be maintained. Additionally, wet/dry cycles experienced in many outdoor exposures mean that anolyte solutions change concentration and form thin films during wetting/drying cycles, meaning that they may move in and out of this concentration range during evaporation cycles [46,123–126]. These types of conditions are not often considered when designing experiments for inhibitor testing, but would provide a much richer set of data for ML.

## 5. Machine Learning

The previous sections explored the scope of potential structures that may need corrosion protection during their lifetime, as well as the large variety of potential electrolytes in which corrosion can occur within these structures. Traditional approaches to testing inhibitors for their degree of protection in these environments are very limited and usually

deconstructed into much simpler tests where only a single variable is examined at one time. This approach is very time-consuming and inefficient. New approaches are being sought, with ML being a very promising contender.

ML is a suite of computational methods that are loosely modelled on how biological brains are thought to function. These algorithms are platform technologies that can be used to find useful patterns in very large and complex data sets without hard-coding the rules or needing to make assumptions about mechanisms. The latter is also a disadvantage, as many ML models are seen as opaque "black boxes" with little or no mechanistic insight. This view is changing, as new algorithms have appeared that can be interrogated, and more chemically interpretable descriptors are being adopted [127]. The main steps in building any sort of ML model are the same. First, a chemically diverse and accurate data set must be obtained. It is very common that the number of data points is relatively small; therefore, methods such as active learning can be invaluable for suggesting the minimal number of new experiments required to substantially improve a model [128].

Ideally, inhibitor values should be normally distributed but this is rarely the case. Transformation of the inhibition percentage data using a logit transfer or similar will help; logarithmic transformation (i.e., conversion of, for example, corrosion current to a logarithmic scale) of data spanning two or more orders or magnitude is also useful, but ideally, inhibition values should be in the form of the concentration at which 50% or 90% inhibition occurs. Sometimes it is sufficient to convert experimental inhibition data into two or more classes (e.g., inhibitor/non-inhibitor). Second, the inhibitors in the training set (a set of performance data, such as the inhibition percentage, used to develop the ML model) need to be converted into numerical descriptors (e.g., Table 2, which shows some molecular descriptors for QM and ML) that capture relevant physicochemical or provenance information. Provenance information is any information about the inhibitor or experimental detail that can be quantified. This data extends the information about the inhibition experiment beyond just the performance data. There are numerous ways of generating molecular descriptors; however, as discussed above, the current focus is on those that can be chemically interpreted to help design new and improved inhibitors. Provenance data or information on the environment can also be added here. This may include details of the impurities in the metal to be protected, the nature of the delivery system, the presence of other molecules or ions, and the pH. This has rarely been done but is an important consideration for the future. If done properly, provenance data may help design inhibitors that are compatible with delivery systems and are more specifically designed for specific environments or application areas. Third, the potentially very large number of descriptors must be reduced to better match the number of training examples to avoid overfitting models and to avoid the presence of many poor descriptors from compromising the accuracy and utility of the model. This feature selection step is very important and is best undertaken using methods that induce sparsity (lots of zeros in data), such as LASSO or MLR with Bayesian expectation maximisation [129]. Fourth, the sparse set of relevant descriptors and the inhibition data are used to train an ML model. The most important choice of algorithm is between linear methods (e.g., multiple linear regression and PLS) and nonlinear methods, such as neural networks, decision trees, and relevance vector machines. The the main determinants of model quality and predictivity are the data and descriptors, most nonlinear ML methods will give similar results when trained on the same training data. Finally, the quality, predictive power, and utility of the model must be assessed. If training data are limited, then cross-validation methods (in which one or more data points are removed in turn, and their properties predicted by a model training on the remaining data) can be useful. If sufficient data exist, a better estimate of model quality is achieved by partitioning part of the data set into a training set used to build a model, and a test set, which is never used in training, to estimate the model's predictive power. A range of statistical metrics can be used to quantify how well models predict the properties of molecules in the training and test sets, with the common ones being $r^2$, RMSE, and MAE. When classification models are generated, traditional metrics from truth tables,

such as accuracy and specificity, can be used, although if the training set is unbalanced (one class has more data points than the other), unbiased metrics, such as F-score or g-means, should be used. Ultimately, the gold standard is to predict the properties of new inhibitors, synthesise them, and test how accurate the predictions are.

While ML has provided some impressive successes in modelling small-molecule corrosion inhibitors (reviewed recently by Winkler [130]), it is currently hampered by the amount of available data and the lack of metadata relating to corrosion environments. If the latter metadata can be collected, as alluded to in the above sections, it should be possible to generate ML models of the performance of corrosion inhibitors that correlate much better with real-world situations. Thus, it is essential to collect information on the performance of more inhibitors and on the physicochemical environments in which they operate. For example, information on the impurities in alloys and their distribution at surfaces could be used if the relevant metadata relating to the alloys can be captured. Likewise, if operational configurations and environments can be encoded adequately by mathematical features, these can certainly be added to the training sets and should improve ML model performance.

As ML methods are universal approximators, they can capture a wide range of complex phenomena apart from the relationships between inhibitor properties and their ability to block corrosion [131]. They may be useful for predicting the propensity of molecules to form barrier coatings via self-assembly or otherwise if sufficient training data could be generated using AFM or computational simulations. ML models could leverage these expensive data to larger areas of chemical space cost-effectively. They could also predict the interaction of inhibitors with coating formulations, and the leaching of inhibitors from relevant coating delivery systems, again if sufficient training data can be obtained. These are just a few corrosion-related research questions that could potentially be answered (or at least the related research could be accelerated) using ML methods. Newer ML methods, such as active learning, can also assist with the data paucity issue [128]. Active learning is a kind of adaptive experimental design that aims to generate the maximum improvement in ML accuracy and domain of applicability with the minimum number of new experiments. This technique has great potential for improving ML models of materials properties that are relevant to corrosion control.

## 6. Conclusions

In this paper, an overview of developments in corrosion inhibitor research focussing on how corrosion performance can be incorporated into machine learning is presented. The authors highlighted different types of environments where corrosion inhibitors are expected to provide protection from corrosion for an underlying metal. These include a range of different structures, as well as underfilm performance. It has also highlighted how localised corrosion contributes to developing electrolytes specific to cathodes and anodes. It has highlighted how the electrolyte within these environments often differs significantly from the standard electrolytes used for inhibitor testing, therefore pointing to the need to do testing of a broader range of electrolytes. One remedy to developing extended databases that incorporate inhibitor performance in these electrolytes is through high-throughput performance testing, which can generate a large amount of data for ML purposes quickly.

**Author Contributions:** In terms of conceptualisation, this paper was based on a presentation made by A.E.H. at the Second Corrosion and Materials Degradation Web conference. All other authors contributed to the writing, editing, and reviewing of the manuscript. All authors have read and agreed to the published version of the manuscript.

**Funding:** This research received no external funding.

**Data Availability Statement:** Not applicable.

**Acknowledgments:** The authors would like to acknowledge the valuable contribution made by George Thompson of the University of Manchester in the area of the X-ray CT work performed in the coatings.

**Conflicts of Interest:** The authors declare no conflict of interest.

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
