# Peer review of "Corrosion Inhibition, Inhibitor Environments, and the Role of Machine Learning"

_cmd, doi:10.3390/cmd3040037_

Round 1
Reviewer 1 Report
This paper presents a substantial review concerning the testing and use of corrosion inhibitors, focusing in particular on some challenges arising from the difference between laboratory testing conditions and the ‘real’ conditions in practice, where complex geometries can lead to correspondingly complex local environments. Partly in response to this challenge, the paper suggests that high throughput performance testing (HPT) in combination with machine learning (ML) techniques may present an improved strategy for future screening and qualification of inhibitors. Overall, the paper contains a substantial amount of information, provides an extensive reference list and clearly outlines the challenges and opportunities in several specific applications. However, I do have a couple of suggestions that I hope the authors will address in a revised version.
(i) The paper is strongest on inhibitors that are contained within coating systems, and most of the discussion concerns these cases. There is also some discussion about inhibitors for use in oil and gas production, especially in relation to Under Deposit Corrosion, but this is much weaker. For example, there are several ASTM standards for evaluating oilfield corrosion inhibitors in the laboratory (e.g. G170, G184, G202, G208) and many NACE papers that discuss results from this type of testing, but none of this is covered. There is also nothing about water treatment for cooling water systems or boilers – both of which have their own extensive literature. I do not see this shortfall as a reason to reject the paper, but even without significant new discussion I think that some improvement could be gained just by providing a little more clarity early on about the different industries and inhibitor types that are being reviewed.
(ii) The shift into the machine learning section is rather abrupt and immediately wades into discussion about “inhibitor values”, ‘logarithmic transformations”, “training sets”, “molecular descriptors” and “provenance data” without first explaining any of these terms. I would suggest that this needs to be remedied before publication.
Author Response
Dear Reviewer,
Many thanks for your time and effort in reviewing the paper. Our response is in the attached document.
Best regards
Tony

Reviewer 2 Report
The manuscript of Hughes et al. reviews recent developments in corrosion inhibition, focusing on how corrosion performance data can be incorporated into machine learning schemes. This is an excellent manuscript that I enjoyed reading it. I believe it will be a useful reference for those interested in machine learning and corrosion inhibition. I am very much in favor of it being published.
I only have several minor suggestions, most of them being typo corrections.
* citations should follow "text [citation]" style but quite often the space is missing (e.g., lines 66, 88, 474, 571, 575, 595, 628, 636)
* all acronyms should be explained when they first appear, i.e.:
- CT is explained on l.500, but it first appears on l.207
- explain "cc" in the μM/cc unit (l.215)
- l.555 & l.556: what is the "M/l" unit; did you mean M or mol/l instead?
* Figures 3: (b) and (c) plots are too small and text/letters cannot be read
* l.256: add space in "(e)and"
* l.317/318: "as summarised in this below"; "in this" appears redundant
* l.319: correct to "derived FROM localised"
* l.323: correct ")]" to ")"
* l.372/373: this appears as in-text reference. Try to convert it into a numeric reference.
* l.381: remove space in ") , with"
* l.382: correct to "AA2024"
* l.384: cation missing in "and (NO3)3"
* l.435: double-space
* l.436: delete space in "ML ."
* Figure 5: upper-left text in the figure is cropped; some letters in the figure are too small (very difficult to read)
* l.451: add space in "5:Schematic"
* l.455: use only commas or parentheses but not both around Figure 6(a)
* Figure 6: the left blue text is chopped
* l.494: delete full-stop in "(a) . Galvanic"
* l.495: delete space in "brine- CO2"
* l.496: add space in "30ppm"
* l.512: correct to "X-ray CT"; correct to "FeCl3·6H2O", i.e., use centered-dot
* l.515: correct to: "(b) Rotated"
* l.518: correct to: "(c) Further"
* l.655: the following text seems to be a leftover: "The following statements should be used “Conceptualization,"
Author Response
Dear Reviewer,
Please see our response to your review in the attached document. Many thanks for the time and care you have taken in reviewing our paper.
Best regards
Tony

Round 2
Reviewer 1 Report
No further comments,